# Safety Climate and Occupational Injuries in the Iron and Steel Industries in Tanzania

**DOI:** 10.3390/ijerph22091372

**Published:** 2025-08-31

**Authors:** Saumu Shabani, Bente Elisabeth Moen, Teferi Abegaz, Simon Henry Mamuya

**Affiliations:** 1Department of Environmental and Occupational Health, School of Public Health and Social Sciences, Muhimbili University of Health and Allied Sciences, Dar es Salaam 11103, Tanzania; saumuskabelwa@gmail.com (S.S.); mamuyasimon2@gmail.com (S.H.M.); 2Department of Global Public Health and Primary Care, Centre for International Health, University of Bergen, 5020 Bergen, Norway; 3Department of Preventive Medicine, School of Public Health, College of Health Sciences, Addis Ababa University, Addis Ababa P.O. Box 9086, Ethiopia; teferiabegaz@gmail.com

**Keywords:** iron and steel industries, occupational injury, safety climate

## Abstract

The iron and steel industries globally have a high prevalence of occupational injuries, which need to be reduced. Obtaining safety climate information from workers assists in understanding the safety status at the workplace. This study aimed to assess the safety climate in the iron and steel industries and its association with occupational injuries. A cross-sectional study was conducted in four iron and steel industrial sites in Tanzania. Three hundred and twenty-one workers from the production lines and 50 managers/supervisors participated. The data were collected by interviews using the Kiswahili version of the Nordic Safety Climate Questionnaire (NOSACQ-50) and the modified International Labor Organization (ILO) manual on methods for occupational injuries. The managers/supervisors scored higher than the workers in five of the NOSACQ-50 dimensions. Most workers with low scores on safety climate had experienced occupational injuries. Analyses of the workers who had been injured at work showed that the dimensions ‘management safety priority and ability’, and ‘management safety justice’ were significant predictors of occupational injuries in the iron and steel industries, when adjusting for working years and working hours per day. This indicates that safety climate is a predictor of occupational injuries, and it is important to improve the safety priorities and commitments among the employees.

## 1. Introduction

The iron and steel industries produce iron and steel, essential materials for construction and manufacturing. Steel is an alloy made from iron and carbon, known for its strength and durability [1]. Steel production involves separating metals from iron ore and/or scrap and subsequently refining the metal to a purer form. The steel-making has two stages. First, the reduction stage, i.e. the iron-making stage, where iron ore is reduced to hot metal. The second stage is the oxidation, i.e., the steel-making stage, where hot metal is refined to steel [1]. Tanzanian steel industries rely on imported billets and metal scraps for steel-making. The sector plays an important role in the production of essential materials for other industries, such as the construction and automobile industries, and is important in the development of low- and middle-income countries. However, the iron and steel industries are among the most dangerous workplaces in the world, compared to other manufacturing industries [2,3]. Workers in the iron and steel industries are at risk of various injuries because of the use of mechanical equipment and energy-intensive processes. There are numerous potential risks of occupational accidents associated with the manufacturing of iron and steel, including high temperatures, working at heights, working in enclosed spaces, working near moving and unsecured machinery, the presence of sharp metal pieces, and heavy lifting. In a systematic review and meta-analysis, the iron and steel industries globally were reported to have a prevalence of occupational injuries of 55%, which is considered very high [4]. In Tanzania, the situation in these industries is even worse; a study from 2025 shows an injury prevalence of 65.1 [5]. The high rate of occupational injuries in these industries is reported to be attributed to poor occupational health and safety (OHS) implementation at these workplaces [4,5,6]. It is uncertain why OHS is not widely implemented in iron and steel companies, even though national laws, regulations, and an ILO code of practice on health and safety emphasize the importance of OHS measures being adopted in these industries to protect workers.

In 2015, the Tanzanian government aimed to be a semi-industrialized nation by 2025 to create job opportunities for young Tanzanians and build a modern economy for the country and its people [7]. This accelerates the evolution of the iron and steel industries in Tanzania, where approximately 14,889 people are currently employed [8], and more jobs are expected to be created; it is crucial to concentrate on this industry and work on preventing occupational injuries. To benefit both the national economy and the iron and steel sectors, conducting a study to improve safety conditions and ultimately protect the workers is essential. To accomplish this, we need to understand the perception of the workers about the available safety policies, procedures, and practices in the iron and steel industries in Tanzania (safety climate). More knowledge of this type will give us a picture of the overall performance of occupational safety in these workplaces.

Safety climate refers to the shared perceptions of employees of how safety is prioritized, valued, and enforced within an organization at a given point in time. Safety climate describes individual perceptions of safety in the work environment [9]. The safety climate can influence an individual’s OHS behavior by motivating them to engage in health and safety practices. To achieve this, the organization needs to have strong management support and commitment to safety, for instance, with an emphasis on worker safety training, regular and open communication, safety promotion, environmental control, good housekeeping, a stable workforce, and employee involvement in health and safety activities and decision-making processes [10]. A positive safety climate seems to lead to a change in employee safety behavior and safety outcomes in the organization [10]. It has been shown that higher compliance with the use of personal protective equipment occurs when workers perceive that there is an atmosphere in the organization that supports safety [11]. Likewise, various studies have demonstrated a relationship between safety climate and occupational injuries [12,13]. A supportive safety climate, where safe behavior is encouraged, seems to be linked to fewer injuries. In contrast, an unsupported safety climate, where safe behavior is not encouraged, seems to be linked to more frequent injuries [14]. 

Managers/supervisors may perceive their safety climate differently. This might be due to their different roles in terms of safety within the workplace. The perception of managers/supervisors about safety can be shaped by knowledge of national safety laws and regulations, safety management goals, the organization’s national safety standard compliance, safety management procedures, and associated safety technology and methods [15]. The management’s positive perception of safety can be the foundation of safety culture within the workplace. A safety-oriented leadership strategy improves compliance and has a favorable effect on how workers perceive and respond to potential hazards in the workplace [16]. If both managers and employees perceive the safety climate positively, this indicates that OHS standards within the workplace are adhered to. Ultimately, this may reduce the rate of occupational injuries [17]. 

Many studies have investigated the safety climate in different types of industries, for example, in agriculture [18,19], oil and power [20,21], nursing homes, healthcare [22,23], and the construction sector [24,25,26]. However, even though the iron and steel industries are recognized as one of the most hazardous workplaces in the world, with a high prevalence of occupational injuries, few studies have examined the relationship between safety climate and occupational injuries in these industries. One survey from Turkey has been performed in the metal industry, indicating an association between safety climate and occupational injuries [27]. A study from Taiwan explored the safety climate, but did not look at the relation to injuries [28]. Both these studies used a survey instrument developed in English, in Western Europe, and the validity of the instrument is questionable. A few other studies on safety have been performed in the iron and steel industry, but these have not focused on safety climate [6,29]. However, a validation study of a safety climate model in the metal processing industry was performed, suggesting a link between safety climate and safety behavior [30]. 

Given the limited and inconclusive research in the field of safety climate and occupational injuries, it is essential to conduct studies using well-established and validated instruments. Occupational injuries represent a significant and serious issue in the iron and steel industries and warrant focused investigation. Conducting such a study in Tanzania is particularly important, as the country may reflect the conditions faced by other low-income nations with similar challenges. This study will enhance understanding of the relationship between safety climate and occupational injuries, thereby contributing to the existing body of literature. The findings will provide valuable insights for policymakers and organizations such as the Tanzanian Occupational Safety and Health Authority, supporting the development of more effective strategies to improve employer compliance with safety regulations and ultimately reduce the incidence of workplace accidents and injuries. Also, it is important to consider that managers and employees may perceive their safety climate differently [31], and it is therefore useful to study both groups in the industries. 

This study aimed to assess the level of safety climate and its association with occupational injuries among workers in the iron and steel industries. The opinions of the employees and their managers have been studied separately. 

## 2. Materials and Methods

### 2.1. Study Design and Population 

We conducted a cross-sectional study in the iron and steel industries in the Dar es Salaam and Pwani areas of Tanzania from July 2022 to September 2023. This industry has many companies, and we selected four of these, without any knowledge about their working conditions and injury prevalence. Two companies from each region were selected from the Tanzanian Occupational Safety and Health Authority (OSHA) registry. This registry includes companies that adhere to the Occupational Health and Safety Act in Tanzania. The sample size was calculated by using OpenEpi, version 3. We used figures from an Ethiopian study, which showed a work-related prevalence of 33% for injuries, a significance level of 0.05, and a statistical power of 95 [6]. The calculated sample size was 381. We were given access to the list of production line workers registered on the payroll from each company and selected the number of workers proportionate to the number of employees. Company A had 125 workers on the production line, and we randomly selected 99 participants from this group. Similarly, company B had 76 production workers, and 60 workers were selected; company C had 116 production workers, and 92 were selected; company D had 165 workers, and 130 were selected. In three companies, we had a response rate of 100%, but in company D, only 70 workers (54%) consented to participate. The criteria for selecting the study population were that participants must have been employed for at least twelve months under the same leadership, present during the study period, and be at least 18 years of age. 

The overall response rate for all workers who consented in the four companies was 84%. For managers/supervisors in these four companies, all who were available during the interview were invited to participate in the study, and 50 managers/supervisors consented and participated. Their response rate was 100%.

### 2.2. Nordic Safety Climate Questionnaire (NOSACQ-50) 

The Nordic Safety Climate Questionnaire is a valid instrument that evaluates individuals’ perceptions of safety climate. The questionnaire was developed by a team of specialists from various Nordic countries in 2011, but has been validated for use in 50 different languages, including Kiswahili. This instrument has been used in several studies [12,14,18,25,27]. According to its validation study, the NOSACO-50 has a very high Item-level Content Validity Index (I-CVI) of >0.76 for all items and scales, as well as a high Scale-level Content Validity Index (S-CVI). This indicates a high level of agreement among experts regarding the scale’s relevance and clarity [32,33]. The NOSACQ-50 consists of 50 questions that assess seven dimensions of safety climate. The seven dimensions include 22 items dealing with three managerial dimensions: ‘management safety priority and ability’ (9 items, of which four are negated or reversed); ‘safety empowerment’ (7 items, 2 negated); and ‘safety justice’ (6 items, 2 negated). The remaining 28 items deal with the worker’s dimensions: ‘safety commitment’ (6 items, 3 negated); ‘safety priority and risk non-acceptance’ (7 items, 6 negated); ‘peer safety communication, learning, and trust in safety ability’ (8 items, 1 negated); and ‘trust in the efficacy of safety systems’ (7 items, 3 negated). As far as the assessment criteria are concerned, a four-step Likert scale is used for rating each statement using the terms strongly disagree, disagree, agree, and strongly agree, which correspond to a 1–4 rating scale in the case of positively formulated statements or 4–1 for the reversed statements. When calculating the scores from the questionnaires, a mean score was calculated for each dimension per participant, whereas the individual mean scores were calculated as the number of positive and reversed responses divided by the number of answered questions. After calculating the individual mean, we then calculated the total mean for the population for each of the 7 dimensions as a ‘mean of the mean’.

Based on the criteria for interpreting safety climate dimension scores, a score greater than 3.30 indicates that the safety climate level of the workplace is good. Safety climate dimension scores between 3 and 3.30 are considered fairly good with a slight need for improvement, scores between 2.99 and 2.70 are considered fairly low with need for improvement, and a score less than 2.7 indicates low with a great need for improvement [32,33]. In our study, we adapted the Kiswahili version of the Nordic Safety Climate Questionnaire (NOSACQ-50), available on the website, and we combined good and fairly good as ‘good score’ and fairly low and low as ‘low score’.

### 2.3. Occupational Injuries

Questions from an ILO manual on methods for collecting occupational injury statistics through household and establishment surveys [34] were used to gather information on occupational injuries. This section of the questionnaire was originally in English and was translated into Kiswahili, then back-translated into English to ensure accuracy. No modifications to the questionnaire were deemed necessary following the translation process.

The primary outcome variable in our analysis was the occurrence of occupational injuries among workers in the iron and steel industries within the past year. Occupational injuries were specifically defined as accidents resulting in injury or disease that occurred at the workplace while the worker was engaged in duties on behalf of the employer [34]. Furthermore, to be classified as an occupational injury, the incident must have led to the worker being incapacitated and unable to work for at least one full day beyond the day of the accident [34].

### 2.4. Pretesting

A pre-test was conducted for the whole questionnaire involving 5% of the total calculated sample size. A pre-test was done in another industry that was not participating in the present study. A few modifications were made afterwards, as some questions needed more words to be clearly understood.

### 2.5. Data Analysis

The collected data were cleaned, coded, entered, and analyzed using the IBM Statistical Package for Social Sciences (SPSS), version 23, IBM Corp, Armonk, NY, USA. Continuous variables were described by mean and standard deviation (mean; SD), and categorical variables were described by proportion (%). Chi-square tests and Fisher’s exact tests were used to assess whether the demographic variables differ significantly between the two groups. To compare the scores of safety climate dimensions between managers/supervisors and workers, a Mann–Whitney test was used because the variables (dimensions) were not normally distributed. Multivariate logistic regression analyses were performed by including variables with a p-value of less than 0.25 from univariate regression analyses. Statistical significance was defined as a p-value less than or equal to 0.05.

## 3. Results

### 3.1. Description of Study Participants: Managers and Workers

A total of 371 participated in this study, 321 production workers with a response rate of 84%, and 50 managers/supervisors with a response rate of 100%. In the production line, all the participants were men, while among managers/supervisors, 48 were male and 2 were female. The mean age of the participants was 32 (SD = 8) for workers and 30 (SD = 5) for managers/supervisors. Many participants had fewer than or equal to 4 working years, with 76% and 56% for workers and managers/supervisors, respectively. Eighty percent of the managers/supervisors worked more than ten hours per day, and the figure for the workers was 76% (Table 1).

### 3.2. Safety Climate Level Among Managers and Workers

The results showed that managers/supervisors have higher mean ranks than the workers on the Nordic Safety Climate Questionnaire dimensions of ‘management safety empowerment’, ‘management safety justice’, ‘workers’ safety priority and risk non-acceptance’, and ‘workers’ trust in the efficacy of safety systems’ (Table 2).

### 3.3. Prevalence of Occupational Injuries

The prevalence of occupational injuries was 65.1% per year. The occupational injuries were dominated by superficial injuries (abrasions, blisters, contusions, puncture wounds) experienced (110/52.6%), followed by dislocation, sprain, and strain injuries, which were experienced by 13.9%. Burns and corrosion injuries were experienced by 12.9% and cuts by 11% (Figure 1). More details on the injuries can be found in a previous publication [5].

### 3.4. Safety Climate and Occupational Injuries Among Workers

The results on safety climate and occupational injuries revealed that the majority of workers who reported being injured at work had low safety climate scores. Whereas in dimension one, 80.4% of workers who reported being injured scored low, 59.4% of injured workers scored fairly low, 32.7% scored fairly good, and 27.3% scored good, in dimension two, 70.7% of injured workers scored low, 73.3% of injured workers scored fairly low, and 63.7% and 42.6% scored fairly good and good, respectively, as shown in Appendix A.

### 3.5. Association Between Safety Climate Score and Occupational Injuries in the Past Year Among Workers in the Production Line of the Iron and Steel Industries

The results show that the scores for the dimensions ‘management safety priority and ability’ and ‘management safety justice’ were both significant predictors of occupational injuries in the iron and steel industries (Table 3) in multivariate regression analyses, adjusting for work experience and working hours.

## 4. Discussion

The findings indicate that the safety climate level among workers in the iron and steel industries is low. Managers/supervisors reported higher safety climate scores than workers, suggesting a poor safety culture at these workplaces. Workers with lower safety climate scores were more likely to have experienced occupational injuries compared to those with higher scores. Specifically, the dimensions ‘management safety priority and ability’, and ‘management safety justice’ emerged as significant predictors of occupational injuries in the iron and steel industries. 

These results are consistent with previous research, including a study conducted at twin plants in Denmark, where workers at plant B—who had high accident rates—also scored lower across nearly all safety climate dimensions [35]. Similar patterns were observed in a study from a metallurgical enterprise in Poland, which found significantly higher safety culture among workers in the ‘no accident’ group compared to the ‘accident group’ [36]. 

In our study, low scores on the ‘management safety priority and ability’ and low scores on ‘management safety justice’ were significantly associated with increased risk of occupational injuries. This aligns with findings from a study in Korea, which reported that occupational injury rates were more than twice as high in workplaces with unfavorable safety climates, compared to those with favorable ones [13]. Notably, factors such as “not encouraging employees to follow safety rules when on a tight schedule” were associated with a high prevalence of occupational injuries [13]. 

These findings suggest that management in the iron and steel industries may prioritize production over safety, reflecting a lack of genuine commitment to occupational health and safety. Although managers may be aware of the importance of health and safety regulations and the potential consequences of neglecting them, their actions do not reflect this awareness. Instead, safety measures are perceived as costs rather than investments, leading to a focus on short-term financial gains at the expense of long-term benefits associated with a safe work environment. 

A low score on the ‘management safety justice’ dimension was also a significant predictor of occupational injuries. This finding is consistent with a study conducted in the construction industry in Taiwan, which reported a higher risk of occupational accidents among men who perceived low levels of workplace justice [37]. Likewise, a study in South Korea showed that drivers with low levels of perceived workplace justice were significantly more likely to be involved in traffic accidents [38]. ‘Management safety justice’ is a dimension of the safety culture that reflects the workers’ perceptions of how the management at the workplace responds to accidents and incidents in the workplace. In our study, low scores on this dimension suggest that the workers felt management did not handle workplace accidents properly or fairly. Such perceptions may discourage employees from reporting incidents due to fear of blame or punishment. Ultimately, this can hinder the identification of causes and implementation of corrective measures, which can increase the risk of recurring occupational injuries. The present study showed that the workers scored lower than their managers/supervisors in five dimensions of the safety climate instrument. The lower perception of the safety climate among workers might be due to the difference in roles and responsibilities when comparing workers and managers/supervisors. The different perceptions of safety in the organization may hinder the implementation of occupational health and safety to mitigate workplace hazards. Better perceptions of safety climate dimensions among managers than among workers have also been observed in another study conducted in the United States. This study involved 1831 truck drivers and their 219 supervisors and found that supervisors provided higher ratings for the safety climate scales ‘management safety empowerment’ and ‘management safety justice’ [31]. These differences in safety perceptions can be explained because managers might want to show how well they perform on safety issues in the workplace for the sake of enhancing the company’s image.

The managers/supervisors scored low on the dimension of management safety priority and ability. This finding is contrary to studies done in Sweden and Colombia [39,40], where the managers/supervisors scored higher on the dimension of management safety priority and ability. This difference can be explained by the fact that the managers/supervisors always want to show a good image of their companies. Our findings indicate that organizations were not prioritizing safety, which, in turn, may affect the whole hierarchical structure of the company. The priorities of the organization are largely communicated by the managers/supervisors to their workers. If managers are perceived not to be committed to safety and not to prioritize safety over other goals, this practice could be transferred to their subordinates, and ultimately, unsafe behavior would be expected [25,31]. Likewise, in the worker safety commitment dimension, the managers/supervisors had a lower score than workers, and their differences were statistically significant. This finding is different from the study conducted in Thailand among production line workers in food manufacturing. The low score among managers in our findings tends to portray different safety cultures available within the workplace as a result of poor leadership and failed role modeling of managers/supervisors in terms of safety practices.

Otherwise, the dimension of safety priority and risk non-acceptance had the lowest score for both managers/supervisors and workers. This result is consistent with a study conducted in the construction industry in Colombia, where the findings showed that the dimension of workers’ safety priority and risk non-acceptance had the lowest safety climate score across the three groups studied: workers, managers, and supervisors [41]. These results are serious and show the need to improve safety. 

Generally, the results support the notion that safety outcomes in the workplace depend on how workers perceive an organization in terms of safety policies, rules, and practices. If the workers’ perception is negative, it will automatically encourage unsafe behavior, which is associated with injury occurrence. Our findings tell us that management involvement in safety is an important factor in occupational injury occurrence at the workplace [33,34]. A more positive safety climate in the workplace could result in a reduction in injuries, which would be advantageous for the organization. This may lead to improved health and performance among the employees, lower compensation costs, fewer employee turnovers, lower insurance premium costs, and fewer working hours lost. To improve workplace safety in the context of this study, management must demonstrate and maintain integrity in health and safety leadership. Safety must be prioritized over production to ensure mutual and long-term benefits for both the industry and its workers. Management should actively empower employees in all matters related to occupational health and safety, fostering a sense of inclusion and ownership. This approach is likely to enhance adherence to occupational health and safety (OHS) policies, procedures, and practices, thereby contributing to accident prevention. Comprehensive safety education for both management and workers is warranted, along with support from relevant authorities, to facilitate the successful implementation of these measures.

### The Strengths and Limitations of the Study

This is the first study conducted in Tanzania assessing the safety climate and occupational injuries within the iron and steel industries. The response rate in this study was high, as the workers were generally willing to provide information. However, the study has some limitations. Because a cross-sectional design was employed, causal relationships between safety climate and occupational injuries cannot be established. It remains unclear whether experiencing injuries influences perceptions of an inadequate safety climate or whether a poor safety climate contributes to the occurrence of injuries. Nonetheless, the findings clearly indicate that the safety climate is suboptimal, and the incidence of occupational injuries is high, suggesting a need for targeting safety improvements within these companies. One priority should be to enhance management training and motivation regarding occupational health and safety. Future studies in this area should adopt a longitudinal study design to better capture temporal relationships between safety climate and injury occurrences. 

Additionally, data were collected through self-reports, which may introduce information bias due to recall inaccuracies or misreporting. However, this risk was mitigated by using a validated tool for obtaining injury information and limiting the recall period to the past year. Another potential source of bias is social desirability, as the workers might have hesitated to provide truthful responses due to concern that management would be informed. To address this, interviews were conducted in a private room, and the participants were assured of confidentiality and anonymity in reporting.

The safety climate data were collected using the Kiswahili version of the NOSACQ-50 validated tool to obtain data on safety climate, which supports the validity of our findings. The study was conducted in Tanzania, including four factories, from two different regions. It is likely that these factories are representative of the iron and steel industries in the country, although this cannot be confirmed with certainty. Future studies should be larger and include more factories, and international studies would be of great interest. However, one should be careful when generalizing the findings to other countries due to the variation in occupational health and safety policies, regulations, and practices. Nevertheless, similar findings might be observed in similar factory sites in other low- and middle-income countries.

## 5. Conclusions

This study showed low safety climate scores among workers in the iron and steel industries. Managers/supervisors scored higher than the workers. The workers with low scores regarding safety climate were more likely to have experienced occupational injuries compared to those with higher scores. ‘Management safety priority and ability’ and ‘management safety justice’ were both significant predictors of occupational injuries. These findings demonstrate the need for safety improvement in this industry. The best approach to achieve this is through workers’ involvement in all issues regarding workplace safety, such as engaging them in safety discussions and decision-making processes to enhance the safety culture.

## Figures and Tables

**Figure 1 ijerph-22-01372-f001:**
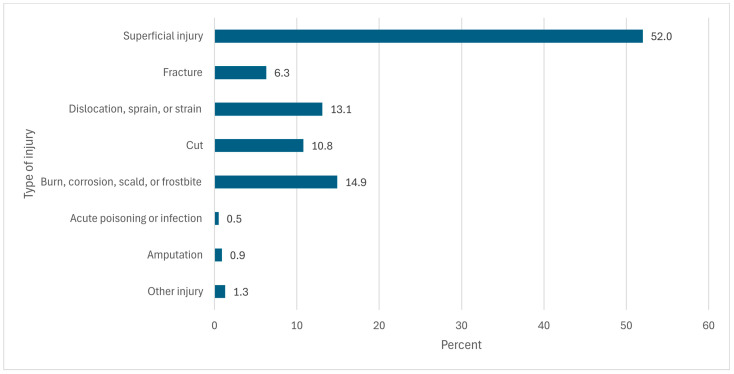
Types (in percentage) of occupational injuries among workers in the iron and steel industries.

**Table 1 ijerph-22-01372-t001:** Socio-demographic characteristics among the participants from the iron- and steel- industry: 371 workers and 50 managers/supervisors.

Variable	Workers	Managers/Supervisors	*p*-Value
	No (%)	No (%)	
Age group			0.121 ^2^
18–30	179 (56)	31 (62)	
31–44	107 (33)	18 (36)	
≥45	35 (11)	1 (2)	
Education			<0.001 ^2^
Never been in school	19 (6)	-	
Primary school	133 (41)	3 (6)	
Secondary school and above	169 (53)	47 (94)	
Marital status			0.134 ^1^
Never married	83 (26)	18 (36)	
Ever married	238 (74)	32 (64)	
Factory			0.340
A	96 (31)	12 (24)	
B	60 (19)	15 (30)	
C	92 (29)	13 (26)	
D	70 (22)	10 (20)	
Section			
Furnace	139 (43)	-	-
Rolling mill	182 (57)		
Working years			0.003 ^1^
1–4 years	245 (76)	28 (56)	
>4 years	76 (24)	22 (44)	
Working hours per day			0.337 ^1^
≤10 h	77 (24)	10 (20)	
>10 h	244 (76)	40 (80)	

^1^ Chi-square test. ^2^ Fisher’s exact test.

**Table 2 ijerph-22-01372-t002:** A comparison of answers from workers and managers/supervisors of the mean scores on 7 Nordic Safety Climate dimensions.

Dimension	Managers or Supervisors Mean Rank	Workers Mean Rank	Mann– Whitney	*p*-Value ^1^
Management safety priority and ability	165.73	189.16	7011.500	0.150
Management safety empowerment	251.46	175.80	11,298.000	<0.001
Management safety justice	265.55	173.61	12,002.500	<0.001
Worker safety commitment	41.12	208.57	781.000	<0.001
Workers’ safety priority and risk non-acceptance	219.88	180.72	9719.000	0.016
Peer safety communication, learning, and trust in safety ability	112.84	197.40	4367.000	<0.001
Workers’ trust in the efficacy of safety systems	232.79	178.71	10,364.500	0.001

^1^ Mann–Whitney test.

**Table 3 ijerph-22-01372-t003:** Association between safety climate score and occupational injuries in the past year among workers in the production line of the iron and steel industries: 209 who had experienced an injury, and 112 who had not.

Variables	Occupational Injury		Univariate COR (95% CI)	Multivariate AOR (95% CI)
	Yes	No		
Management safety priority and ability				
Good	26 (27.3)	59 (72.7)	1	1
Low	183 (80.4)	53 (19.6)	7.835 (4.51–13.63) *	6.73 (3.65–12.41) *
Management safety empowerment				
Good	78 (42.5)	60 (57.5)	1	1
Low	131 (70.7)	52 (29.3)	1.94 (1.22–3.09) *	0.98 (0.55–1.75)
Management safety justice				
Good	51 (29.6)	56 (70.4)	1	1
Low	158 (92.3)	56 (7.7)	3.09 (1.90–5.04) *	1.80 (1.02–3.19) *
Worker safety commitment				
Good	84 (57.1)	45 (42.9)	1	-
Low	125 (68.1)	67 (31.9)	0.999 (0.626–1.596)	-
Workers’ safety priority and risk non-acceptance				
Good	16 (57.1)	12 (42.9)	1	-
Low	193 (65.9)	100 (34.1)	1.447 (0.659–3.178)	-
Peer safety communication, learning, and trust in safety ability				
Good	171 (29.6)	83 (70.4)	1	1
Low	38 (92.3)	29 (7.7)	0.64 (0.37–1.10)	0.666 (0.35–1.25)
Workers’ trust in the efficacy of safety systems				
Good	154 (65)	83 (35)	1	-
Low	55 (65.5)	29 (34.5)	1.022 (0.606–1.724)	-
Working years (Experience)				
1–4 years	169 (69)	76 (31)	2.001 (1.183–3.384)	
>4 years	40 (52.3)	36 (47.4)	1	
Working hours per day				
≤10 h	38 (49.4)	39 (50.6)	1	
>10 h	171 (70.1)	73 (29.9)	2.404 (1.423–4.060)	

* Adjusted for working years and working hours. COR = crude odds ratio, AOR = adjusted odds ratio, 95% CI = 95% confidence interval.

## Data Availability

Data for this study are available upon reasonable request from the corresponding author.

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
