# Peer review of "Safety Climate and Occupational Injuries in the Iron and Steel Industries in Tanzania"

_ijerph, 2025, doi:10.3390/ijerph22091372_

Round 1
Reviewer 1 Report
Comments and Suggestions for Authors
1 In the Introduction, the authors do not elaborate on the necessity of the study. What are the disadvantages and limitations of current studies? The authors emphasize in the fourth paragraph that iron and steel industry has not paid attention to the safety climate, which is not comprehensive. Some studies have already conducted research on the safety climate in iron and steel industry, although it is not referred to as safety climate in their papers. For example, Work related injuries and associated risk factors among iron and steel industries workers in Addis Ababa, Ethiopia, and Designing new environmental policy instruments to promote the sustainable development of iron and steel production in China: A comparative analysis of cleaner production assessment indicator systems and the assessment principles of the green factory. Disclaimer: These two papers are not related to the reviewer. Authors should point out the limitations of current studies based on literature review on safety climate in iron and steel industry. Give concise and clear presentation in Introduction.
2 The application of research findings should be increased. This study presents a low safety climate score among workers in the iron and steel industry. So, how to improve the safety climate score among workers in the iron and steel industry? Suggestions like this should be added in the manuscript.
Author Response
A file is enclosed with our responses to Reviewer 1.

Reviewer 2 Report
Comments and Suggestions for Authors
1- Topic is interesting, but this paper need to improve.
2-Please elaborate on the managerial insights of the present study in a distinct section
3-The introduction has been revised (focusing on the background of research, the motivation of the research, and the novelty) and the literature review.
4- Your methodology is too brief and unclear
5- What is the criteria inclusion?
6-CVI, CVR indices of the Nordic security air questionnaire (NOSACQ-50) should be reported.
7-Figure 1 is not suitable for the present journal, the authors captured the graphic. Figure 1 should be improved.
8-There are some punctuation issues. The paper needs to be double checked in this regard
9-The Conclusion section is very poorly written and should be revised
10-The literature review should be enriched to be more up-to-date and more comprehensive
Author Response
A file is enclosed with our responses to reviewer 2.

Reviewer 3 Report
Comments and Suggestions for Authors
The subject of this manuscript aligns well with the scope of IJERPH, as it investigates the relationship between safety climate and occupational injuries in iron- and steel- industries in Tanzania. The authors employ a cross-sectional survey using a validated instrument, the Nordic Safety Climate Questionnaire, and the subsequent statistical analysis is clearly presented, with its limitations appropriately acknowledged.
The findings are compared to those of similar studies, which enhances the relevance of the results. However, the Discussion section could be improved by focusing more on this comparative analysis. Overall, the manuscript is well written, though it contains a few minor grammatical and syntactical errors that should be corrected.
My recommendation is to accept the submitted manuscript with minor revisions, as outlined in the following comments.
Lines 58-63. “Recently…worldwide” This paragraph is very vague. The authors should specify their data and dates.
Lines 63-68. These generic statements could be replaced by specific arguments on the aim and scope of this research in combination with the last paragraph of the Introduction section.
Lines 94-95. “[…] no research…industries” Although scarce, there do exist certain similar studies, as it is also evident in the Discussion section. Moreover, certain studies have associated indicators of safety climate with occupational injuries.
Lines 104-105. Some more information on the time and duration of the study could be useful.
Lines 105-106. Are there any data on the representability of the study’s sample? If not, a relative reference should be made in the limitations section.
Line 269. Correct to “in” five dimensions.
Line 278. Correct to “can be” explained.
Line 285. Replace to word “habit”
Lines 296, 301. Correct punctuation.
Author Response
A file is enclosed with reponses to the comments from Reviewer 3.

Reviewer 4 Report
Comments and Suggestions for Authors
This manuscript investigates the relationship between safety climate and occupational injuries in Tanzanian iron and steel industries using the NOSACQ-50 instrument. The research addresses an important gap in occupational health literature for this high-risk industrial sector in a low-to-middle income country context..
Recommendation: Accept with minor revisions.
Was used appropriate methods such as: Mann-Whitney U test for non-normally distributed safety climate dimensions, Multivariate logistic regression with p<0.25 inclusion criteria, Adjustment for confounding variables (working years, working hours).
Methodological Constraints: Recall Bias: Self-reported injury data over 12-month recall period may introduce information bias. Selection Bias: OSHA-registered companies may not represent industry-wide safety standards. Social Desirability Bias: Workers may provide responses they perceive as favorable to management.
Recommendations:
- Enhance generalizability discussion for other low-to-middle income countries.
This study provides valuable empirical evidence linking safety climate dimensions to occupational injury risk in Tanzanian iron and steel industries. The research contributes meaningful insights for occupational health policy and practice. The identification of management safety priority and justice as significant predictors offers targeted intervention opportunities for injury prevention in this high-risk industrial sector.
Round 2
Reviewer 1 Report
Comments and Suggestions for Authors
I have no comments.